# SparseCodeQ: Extreme Sparse Coding Quantization for Large Vision-Language Models

## Abstract

In this paper, we propose an extreme sparse coding quantization framework of 2-bit large vision-language models (LVLMs) for efficient multimodal reasoning. Conventional codebook-based quantization methods assign the same codeword number to all weights ignoring the significant variance of weight salience, which leads to substantial discretization errors. On the contrary, we flexibly assign optimal codeword combination for each weight based on weight salience to mitigate the performance degradation with negligible complexity overhead. Specifically, we first select the number of codewords for all weights based on the salience evaluation with second-order information. We then propose hierarchical codeword selection to efficiently search the appropriate codeword combinations from the extremely large codebook for optimal sparse representation. The high-level candidate search selects representative codeword subsets with minimal quantization errors, through which the low-level subset refinement discovers the optimal fine-grained codeword combination for all weights. Finally, we optimize the visual encoder to concentrate the weight salience distribution, which reduces computational overhead because of the decreased codewords for aggregated salient weights. Experimental results demonstrate that our method achieves a 5.58× reduction in model size while outperforming state-of-the-art model quantization methods by 2.78 in performance on the 13B LLaVA model, achieving a notable margin of improvement while maintaining similar computational costs in LVLM quantization.

## 1 Introduction

The large vision-language models (LVLMs) (Liu et al., 2024a; Li et al., 2023a) have achieved groundbreaking results in a wide variety of multimodal reasoning tasks. Powered by their vast parameter counts and complex architectures, they have demonstrated impressive performance in visual question answering (Wu et al., 2023; Lin et al., 2024a), embodied instruction following (Ahn et al., 2022), and robot navigation (Anderson et al., 2018; Hao et al., 2020). Although these models deliver high performance and generalization capabilities, their extensive computational and memory requirements pose challenges for real-world applications. As a result, there is a pressing need to reduce the complexity of LVLMs, which unlocks their potential for a broader range of applications, including deployment on mobile devices and edge devices.

To solve this problem, various model compression techniques have been proposed to reduce model complexity including pruning (Frantar & Alistarh, 2023; Fang et al., 2024), quantization (Tseng et al., 2024a; Lin et al., 2024b; Shao et al., 2023) and efficient architecture design (Zhou et al., 2024; Chu et al., 2024). Among these methods, quantization reduces the precision of model parameters to decrease memory usage and computational overhead, leading to accelerated inference speed and mitigates energy consumption. Conventional quantization methods have introduced salience-based quantization (Frantar et al., 2022; Huang et al., 2024; Dong et al., 2019) which quantize different model parameters with varying precision levels based on their salience. These approaches help to achieve a better trade-off between model efficiency and accuracy by focusing on the salient parameters, but they encounter substantial discretization errors with uniformly distributed quantization function and severely limited range of weight representation under extreme low bitwidth. Recent advances introduce codebook-based quantization methods (Egiazarian et al., 2024; van Baalen et al., 2024) which capture inter-dimensional correlations among weight groups, providing a more adaptable quantization grid across multiple dimensions. Given the non-uniform distribution of model weights,

these methods are particularly effective in representing complex weight patterns and minimizing redundancy by full-precision codebooks. However, current codebook-based methods assign the same codeword number to all weights ignoring the significant variability in weight salience, which leads to substantial information loss because of the ineffective bitwidth assignment.

In this work, we propose an extreme sparse coding quantization framework for 2-bit LVLM quantization called SparseCodeQ to accelerate large vision-language models for efficient multimodal reasoning. Different from existing codebook-based quantization methods which assign same codeword number to all weights, this framework dynamically assigns multiple number of codewords based on weight salience, and selects optimal linear combination of codewords for each weight to mitigate the information degradation with negligible complexity overhead. To be specific, SparseCodeQ first allocates the number of codewords for each model parameter based on the weight salience, where important weights are represented by more codewords for fine-grained model quantization. To search for optimal sparse linear combinations of codewords from the extremely large codebook, we propose hierarchical codeword selection algorithm with high-level candidate search and low-level subset refinement. The high-level candidate search selects representative

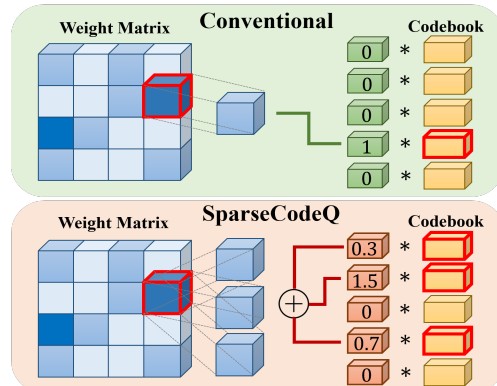

Figure 1: Existing methods assign same codeword number to all weights with significant discretization errors, while we flexibly assign multiple codewords based on weight salience and search the optimal sparse linear combinations of codewords.

codeword subsets with minimal discretization errors, through which the low-level subset refinement discovers the optimal fine-grained codeword combination to represent each weight. We finally optimize the visual encoder to obtain a more concentrated salience distribution, which reduces computational overhead because of the decreased codewords for aggregated salient weight. Experimental results demonstrate that our method achieves a 5.58× reduction in model size while outperforming state-of-the-art model quantization methods by 2.78, with only 2.19 accuracy drop for the FP 13B LLaVA model, achieving a notable margin of improvement while maintaining similar computational costs in LVLM quantization. The main contributions of this work are summarized as follows:

- Our framework is the first method to incorporate sparse coding principles for model compression. This innovative approach enables the efficient representation of non-uniformly distributed model weights and prioritizes the quantization of salient parameters simultaneously.

- We design SparseCodeQ to adaptively assign the number of codewords for each weight and search for the optimal sparse linear combination of fine-grained codewords with minimal discretization errors for all weights, and vision encoder is further optimized with concentrated salience distribution to reduce computational overhead.

- Experimental results demonstrate that our method compresses the model size by 5.58x and outperforms the state-of-the-art quantization methods by 2.78 on 13B LLaVA model while achieving 3.6x memory reduction and 1.3x inference acceleration under extreme low bitwidth.

## 2 RELATED WORK

### 2.1 LARGE VISION-LANGUAGE MODEL

Large vision-language models (LVLMs) have demonstrated outstanding performance mainly due to their rapid adaptation across diverse downstream tasks and strong generalization capabilities. This success can be attributed to the extensive leverage of large-scale image-text datasets (Radford et al., 2021; Jia et al., 2021) and the remarkable generalization abilities of pre-trained large language models (LLMs) (Brown et al., 2020; Touvron et al., 2023). Early efforts integrated extensive commonsense knowledge from LLMs into vision-language representation learning, treating visual information as contextual input for effective utilization of LLMs. For example, Flamingo (Alayrac et al., 2022) employed adapters within LLMs and leveraged a perceiver-like architecture to extract

Figure 2: The overall pipeline of our SparseCodeQ. We select the number of codewords based on weight salience and hierarchically search the optimal sparse linear combination of codewords with minimal quantization errors for all weights. Vision encoder is further optimized with concentrated salience distribution to reduces computational overhead.

visual features for multimodal alignment, achieving high accuracy in few-shot learning for vision-language tasks. Likewise, BLIP (Li et al., 2022; 2023a) used data filtering techniques to boost performance in tasks such as visual question answering (VQA) and image captioning. Although these models demonstrated remarkable vision-language reasoning capabilities, their zero-shot performance remained limited due to the absence of explicit instruction-based training. Despite the significant performance improvements enabled by larger models, the computational demands and storage costs hinder the deployment of LVLMs on resource-constrained devices. To address these limitations, lightweight LVLMs such as TinyGPT-V (Yuan et al., 2023a) and TinyLLaVA (Zhou et al., 2024) have been proposed, focusing on optimizing the architecture by leveraging smaller models for efficient LVLM design. MoE-LLaVA (Lin et al., 2024a) constructs a sparse mixture-of-experts (MoE) model, which selectively activates pathways to simultaneously process image and text features, achieving competitive performance with fewer active parameters. However, the inference costs of these models remain beyond the resource limits of mobile devices or robots due to their low compression ratios.

## 2.2 MODEL QUANTIZATION

Network quantization replaces full-precision tensors with low-precision values and substitutes multiply-accumulate operations with integer arithmetic, thereby substantially reducing both storage and computational costs of neural networks. Conventional quantization-aware training (QAT) methods (Choi et al., 2018; Liu et al., 2020) require fine-tuning network weights using the entire training dataset for rounding, which is often impractical due to the unavailability of data and resources for most users. Recently, post-training quantization (PTQ) (Yang et al., 2024; Yuan et al., 2023b; Huang et al., 2024) minimized the $l_2$ distance between quantized and full-precision tensors to reduce task performance degradation. Zero-shot PTQ further pushes the limits of efficient quantization without relying on real image data. SmoothQuant (Xiao et al., 2023), and ZeroQuant (Yao et al., 2022) addressed activation outliers by removing extreme values through equivalent transformations, facilitating accurate quantization function learning; however, they faced challenges in scaling to very large models due to excessive computational costs. GPTQ (Frantar et al., 2022), AWQ (Lin et al., 2023), and QLoRA (Dettmers et al., 2024) employed low-precision quantization for weight quantization to further reduce computational complexity but failed to achieve effective quantization under extremely low bitwidth.

Moreover, vector quantization (VQ) (Egiazarian et al., 2024; Tseng et al., 2024a) has been explored as an efficient method of weight-only quantization under extremely low bitwidth. These methods represent high-dimensional weight vectors with a set of low-dimensional codebooks which substantially reduce the memory usage while allowing quick reconstruction through simple index references. GPTVQ (van Baalen et al., 2024) and VPTQ (Liu et al., 2024b) leverage second-order optimization to achieve accurate and efficient vector quantization. Nevertheless, these codebook-based quantization methods represent weight groups using a single codeword, where the optimal representation is

optimized by fine-tuning, leading to extreme search cost and suboptimal codeword selection due to overfitting to small calibration sets.

## 3 APPROACH

In this section, we first introduce the preliminaries of quantization and sparse coding for LVLMs and then detail the SparseCodeQ framework which selects the number of codewords and searches the optimal linear combination of codewords with minimal discretization errors, and vision encoder is further optimized with concentrate salience distribution to reduce computational overhead.

### 3.1 SPARSE CODING-BASED QUANTIZATION

Model quantization methods are generally designed to reduce discretization errors between the outputs generated by the original weights $\mathbf{W} \in \mathbb{R}^{d_o \times d_i}$ and the quantized weights $\widehat{\mathbf{W}}$. To maintain efficient memory usage and inference speed, we introduce sparse coding to represent weight groups with a sparse linear combination of codewords. Sparse coding was initially introduced in signal processing to create efficient and interpretable representations of data. In model quantization, our aim is to approximate each weight group by forming a sparse linear combination of codewords from an extremely large codebook $\mathbf{C} = \{\mathbf{c}_i\}_{i=1}^K \in \mathbb{R}^{d_o \times K}$, with $K$ denoting the number of codewords in the codebook. The sparse coding coefficients that represent the contributions of each codeword are denoted by $\mathbf{S} = \{\mathbf{s}_i\}_{i=1}^K \in \mathbb{R}^{K \times d_i}$. The process of deriving sparse coding coefficients is formulated as an optimization problem aimed at achieving a balance between accurately reconstructing the original signal and enforcing the sparsity of the coefficients. This objective $L$ can be formulated as:

$$L = \operatorname*{argmin}_{\{\mathbf{s}_i\}_{i=1}^K} \|\mathbf{W} - \sum_{i=1}^K \mathbf{c}_i \mathbf{s}_i\|_2^2 + \lambda \sum_{i=1}^K \|\mathbf{s}_i\|_0, \tag{1}$$

where $\lambda$ is a regularization parameter that controls the trade-off between reconstruction accuracy and sparsity. Direct optimization of this objective would be NP-hard due to the large number of parameters involved in LVLMs. On the contrary, we propose adaptive codewords selection algoritm to achieve optimal sparse linear combination of codewords for all weights while minimizing discretization errors. Specifically, the optimization problem can be split into two subproblems: (1) determining the optimal allocation of codeword number across different weight groups, and (2) identifying the most representative codewords that facilitate optimal sparse linear combination for all weights. With the optimal trade-off between memory usage and discretization errors, our SparseCodeQ search the optimal linear combination of codewords for each weight.

### 3.2 ADAPTIVE SALIENCE-DRIVEN CODEWORD ALLOCATION

As each weight group varies in importance, determining the optimal allocation of codeword number across different weight groups can effectively balance the trade-off between search complexity and discretization errors. Directly optimizing the sparsity term in equation 1 incurs substantial computational costs due to the exhaustive search space. To mitigate this, we derive an alternative formulation by leveraging the second-order information of the weight parameters to estimate sparsity. For the weight in the loss function, we apply a Taylor expansion as $L(\mathbf{W} + \triangle\mathbf{W}) \approx L(\mathbf{W}) + \frac{1}{2}\triangle\mathbf{W}^T \bigtriangledown^2 L(\mathbf{W}) \triangle\mathbf{W}$, where $\triangle\mathbf{W} = \sum_{i=1}^K \mathbf{c}_i \triangle\mathbf{s}_i$ represents the weight changes after removing some codewords. Due to the fact that the model has already converged on the training dataset, we neglect the first-order term where $\bigtriangledown L(\mathbf{W}) \approx 0$. Consequently, we can quantify the importance $\triangle L_i$ of $i$-th codeword by measuring the error increment in loss function upon its removal, formulated as:

$$\triangle L_i \approx \frac{1}{2}(\mathbf{c}_i \triangle \mathbf{s}_i)^T H_{ii}(\mathbf{c}_i \triangle \mathbf{s}_i), \tag{2}$$

where $\triangle\mathbf{s}_i$ represent the change in the sparse coefficient associated with the $i$-th codeword and $H_{ii}$ denotes the $i$-th diagonal element of the Hessian matrix. Codewords with larger Hessians exert greater impact on overall perturbation of weight representation, while those with smaller Hessians permit safe coefficient pruning with minimal error increments. Therefore, the Hessian matrix can reflect the sensitivity of quantization for each weight, and the sum of this matrix is defined as weight salience.

Specifically, the Hessian matrix $\mathbf{H} \in \mathbb{R}^{d_i \times d_i}$ is computed as $\mathbf{H} = \frac{2}{n} \sum_{i=1}^{n} \mathbf{X}_i \mathbf{X}_i^T$ for the layer-wise reconstruction problem, where $\mathbf{X} \in \mathbb{R}^{d_i \times n}$ represents a $n$ samples calibration set to simulate realistic data distributions. By calculating the weight salience, we can flexibly assign the optimal codeword number for all weights with negligible complexity overhead to reduce information loss.

## 3.3 HIERARCHICAL CODEWORD SELECTION FOR OPTIMAL SPARSE REPRESENTATION

After assigning a flexible codeword number for each weight group, we further identify the most representative codewords that facilitate an optimal sparse linear combination to mitigate the performance degradation. Our goal is to select sparse codewords set $C'$ from an extremely large codebook $C$. Directly optimize sparse coding coefficients in equation 1 would be NP-hard due to the large number of parameters involved in LVLMs. On the contrary, we propose hierarchical codeword search algorithm, which consists of a high-level candidate search to select representative subset followed by a low-level subset refinement for optimal codeword selection. This hierarchical approach effectively overcomes the limitations posed by a large parameter count while generating fine-grained sparse coding coefficients to accurately approximate model weights.

**High-level candidate search.** To deal with the extreme search cost in optimizing total codebook with coefficients for each weight group, we first select the potential codeword subsets with the most similarity to each weight group with fixed coefficients $\mathbf{s}_i$ to reduce the search space. The weight metric $\widehat{\mathbf{W}}$ is appropriately represented with the selected subsets $\mathbf{C}'_s \in \mathbf{C}$ which contains most weight feature while removing most of the redundancy. This subset is composed of a selection of candidate codewords from an extreme large codebook that are particularly effective at reducing discretization error and the similarity $S^k_{\mathbf{c}_i}$ between reconstruction errors $\triangle\widehat{\mathbf{W}}_{k-1}$ of the group weight in $k$-th iteration and the $i$-th codeword $\mathbf{c}_i \in \mathbf{C}$ can be represent with cosine similarity as follows:

$$S^k_{\mathbf{c}_i} = \underset{\mathbf{c}_i \notin \mathbf{C}'_{k-1}}{\operatorname{argmax}} \frac{\mathbf{c}_i \cdot \triangle\widehat{\mathbf{W}}_{k-1}}{\|\mathbf{c}_i\| \left\|\triangle\widehat{\mathbf{W}}_{k-1}\right\|}, \tag{3}$$

where $\|\cdot\|$ represent calculating $L_2$ norm and $\mathbf{C}'_{k-1}$ is the selected codeword subset in $k$-th iteration. After selecting the most representative codeword, the codeword coefficient is further initialized with the optimization of weight reconstruction errors in equation 1:

$$\triangle\widehat{\mathbf{W}}_{k-1} = \underset{\{\mathbf{s}_{k-1}\}_{i=1}^{k-1}}{\operatorname{argmin}} \|\mathbf{W} - \sum_{i=1}^{k-1} \mathbf{c}'_i \mathbf{s}_{i,k-1}\|_2^2, \tag{4}$$

where $\mathbf{s}_{i,k-1}$ represent the coefficients corresponding to $i$-th codewords $\mathbf{c}'_i \in \mathbf{C}'_{k-1}$ in selected $k$-th subset. By iteratively computing the similarity between codewords and the weights, we identify the codeword combination that exhibits the optimal alignment with the weights. With the highest degree of correlation, the selected subsets effectively reduce the search space while maintaining the information gain of the codebook. By focusing on a group of high-quality candidates, this method reduces the quantization error and the candidate search space for subsequent stages to ensure that the algorithm can achieve a more accurate and efficient quantization outcome.

**Low-level subset refinement.** As the previous stage narrows down potential codeword subsets with minimal quantization errors, the low-level subset refinement stage refines these subsets by pinpointing the optimal codeword combinations and further optimizing codewords and sparse coding coefficients. Since the subset remains large, we aim to retain only the small essential codewords while removing the rest. The number of codewords in $\mathbf{C}'$ is determined by salience as described in Section 3.2. To determine which codewords to keep, we iteratively rank them according to their impact on the squared error between the outputs with and without each codeword. We denote the quantized weight matrix that excludes a particular codeword $\mathbf{c}_m$ from the subset $\mathbf{C}'_s$ as $\triangle\widehat{\mathbf{W}}'$. The impact of removing the codeword $\mathbf{c}_m$ can be calculated as:

$$\begin{aligned} \triangle\widehat{\mathbf{W}}'\mathbf{X} &= \|\mathbf{c}_m \mathbf{s}_m \mathbf{X}\|_2^2 \\ &= \langle \mathbf{c}_m \mathbf{s}_m \mathbf{X}\mathbf{X}^T, \mathbf{c}_m \mathbf{s}_m \rangle_F, \end{aligned} \tag{5}$$

where $\langle \cdot, \cdot \rangle_F$ denotes the Frobenius inner product and $\mathbf{s}_m$ denotes the $m$-th element of sparse code coefficients $\mathbf{s}$ for this single quantization group. Therefore, we can progressively reduce the least

---

**Algorithm 1** SparseCodeQ Quantization

---

**Input**: LVLM model, calibration inputs $X$.
**Output**: Quantized LVLM model.
1: Optimize the vision encoder to concentrate the Hessian distributions.
2: **for** block in language model **do**
3:    **for** layer in block **do**
4:       Compute weight salience and initialize codebook $\mathbf{C}$ with K-means.
5:       Perform high-level candidature search to obtain potential codeword subset $\mathbf{C}'_s$ for weights.
6:       Perform low-level subset refinement to refine codeword subsets.
7:    **end for**
8:    Perform block-wise fine-tuning.
9: **end for**

---

influential codewords while optimizing the codeword and corresponding coefficient processes, and re-evaluate the importance of the remaining codewords. As a result, based on the optimization formula in equation 4, we can obtain the optimal codeword combination that minimizes the discretization error. The hierarchical search method we propose significantly reduces the search space for the optimal codeword combination by retaining the majority of weight features and gradually optimizing, resulting in a substantial reduction in discretization error.

### 3.4 OPTIMIZATION VISION ENCODER FOR LVLMS QUANTIZATION

Large Vision-Language Models (LVLMs) employ a vision encoder to convert visual information from image inputs into visual embeddings that are aligned with corresponding text embeddings. Directly deploying optimal codeword combination can reduce discretization errors but increase the memory usage and inference speed due to the increasing codewords. Due to the substantial impact the vision encoder has on the distribution of activations within LVLMs, optimizing the data distribution within the vision encoder can help influence subsequent multimodal layers to achieve optimal trade-off between discretization errors and the search cost.

In sparse coding quantization frameworks, the salience distribution modulates the scale of codeword assignment: weight groups with larger salience adopt more complex assignment strategy, while smaller salience groups are often allocated fewer codewords. Therefore, we aim to concentrate the salience distribution in multimodal layers, allowing salient weight groups with complex codewords to better capture diverse distributions while reducing computational overhead because of the decreased codewords for aggregated weight. However, directly optimizing the salience distribution risk diverging from the pre-trained model with optimization difficulties because numerous layers in LLaMA often introduce conflicting supervision for the visual encoder. On the contrary, we minimize activation entropy of the Hessian matrix in LVLMs to concentrate the salience distribution. Considering that layers contribute differently to the overall loss function, we assign different importance weights to the entropy minimization objectives across layers which are acquired from the matrix traces:

$$L_{ent} = \sum_{k=1}^{n} \sum_{i} \frac{\partial^2 L}{\partial X_i^{(k)2}} \cdot \sum_{j} p(H_{ij}^{(k)}) \log p(H_{ij}^{(k)}), \tag{6}$$

where $X_r^{(k)}$ represents the quantized activation distribution in $k$-th layer. We apply normalization to transform the Hessian into a probability distribution suitable for calculating the information entropy as $p(H_{ij}^{(k)}) = \frac{|H_{ij}^{(k)}|}{\sum |H^{(k)}|}$. The leverage of entropy polarizes the salience distribution. The Hessian traces indicate the influence of the current layer on the quantization errors of the final output (Lopes et al., 2021). Larger trace magnitudes represent a higher influence on the overall discretization errors, and assigning larger weights to those layers can reduce the cross-layer dependency with fast model convergence. Meanwhile, our objective also includes the minimization of discretization errors for the output of the LVLM layers, which can enhance the quantization accuracy for visual representation learning and multi-model reasoning. Finally, the overall objective for visual encoder optimization can be written as follows:

$$L = L_{reg} + L_{ent} + \|\mathbf{W} - \sum_{i=1}^{K} \mathbf{c}_i \mathbf{s}_i\|_2^2, \tag{7}$$

| Method | Bit | SQA | VizWiz | VQA[v2] |
|--------|-----|-----|--------|---------|
| AQLM | | 58.08 | 51.24 | 76.13 |
| SDCA | 2.6 | 59.44 | 51.38 | 76.53 |
| +HCS | | 60.13 | 52.17 | 77.06 |
| Ours | | 60.69 | 52.75 | 77.54 |
| AQLM | | 56.19 | 50.76 | 75.67 |
| SDCA | 2.2 | 57.91 | 50.84 | 76.79 |
| +HCS | | 59.21 | 51.87 | 76.98 |
| Ours | | 59.77 | 52.53 | 77.36 |

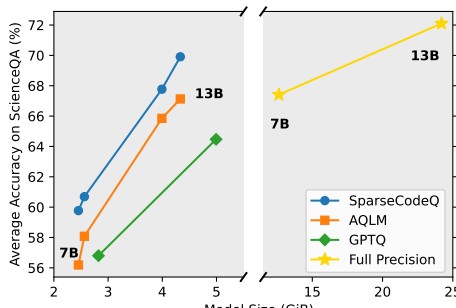

Table 1: Effect of different LVLM quantization method we proposed. We report the accuracy result for 2.6 bits and 2.2 bits of LLaVA-7B model on ScienceQA, VizWiz, and VQA-v2 dataset.

Figure 3: The average answering accuracy on ScienceQA dataset w.r.t. different model size across various quantization methods for LLaVA 7B and 13B models.

where $L_{reg}$ means the auto-regressive loss adopted in training original LVLM to minimize the discrepancy of predicted and target tokens. By optimizing the visual encoder, we can obtain a more concentrated salience distribution to reduces computational overhead because of the decreased codewords for aggregated salient weight.

## 4 EXPERIMENTS

In this section, we conduct extensive experiments for LLaVA benchmarks on various multimodal question-answering datasets including ScienceQA (Lu et al., 2022), VQA-v2 (Goyal et al., 2017), and 6 other datasets to evaluate the effectiveness of our methods. We first conduct ablation studies to evaluate the effectiveness of our SparseCodeQ framework. We then compare our SparseCodeQ with the state-of-the-art quantization methods to show its superiority. Implementation details are introduced in supplementary.

### 4.1 ABLATION STUDY

Since previous codebook-based methods assign the same codeword number to all weights ignoring the significant variance of weight salience, we flexibly assign multiple numbers of codewords and select optimal linear combination of codewords for each weight. We conduct our calibration datasets to fine-tune quantized models and demonstrate the effectiveness of the proposed methods.

**Performance w.r.t. different methods we proposed in question answering process:** In order to investigate the effectiveness of different methods we proposed in SparseCodeQ framework, we conduct an ablation study on various question answering datasets. Table 1 illustrates the answering accuracy for our method under different bitwidth for LLaVA-7B model. As observed in the second rows, the adaptive Salience-Driven Codeword Allocation (SDCA) module plays a critical role in the final performance, as it enables the optimal trade-off between quantization errors and the search cost by allocating an appropriate number of codewords to each weight. However, a fundamental limitation of the SDCA method arises from its static sparse representation of codewords during the fine-tuning while the optimal sparse linear combination for each weight evolves. As a result, SDCA inevitably selects a suboptimal codeword combination that compromises overall performance. On the contrary, our Hierarchical Codeword Selection (HCS) method dynamically selects the optimal sparse linear combination of codewords in a hierarchical manner. The high-level candidate search selects codeword subsets with minimal quantization errors, while the low-level subset refinement discovers the optimal codeword combination during fine-tuning. The fine-grained quantization function search further enhances the performance in the third rows. Moreover, Visual encoder optimization (VEO) also significantly modifies the distribution of the Hessian metrics in LVLMs for fine-grained search space allocation so that precise discretization function can be acquired with further reduction of search cost. We also evaluate the fine-tuned vision encoder with FP language model in LLaVA-7B, demonstrating no obvious improvement without concentrated Hessian distribution (67.41 vs. 67.36).

**Performance w.r.t. different codeword selection methods:** Table 2 shows the bitwidth setting, quantization errors, and accuracy on ScienceQA dataset. We analyze the impact of various codeword

| Method | Bit | Errors | Accuracy |
|--------|-----|--------|----------|
| SDCA+Random | 2.6 | 0.040 | 58.50 |
| SDCA+OMP | 2.6 | 0.038 | 59.44 |
| SDCA+HCS | 2.6 | 0.031 | 60.13 |

| Method | Bit | Memory | Speed | Accuracy |
|--------|-----|--------|-------|----------|
| FP | 16 | 14.4GB | 19.8 tok/s | 67.41 |
| AQLM | 2.6 | 3.4GB | 28.9 tok/s | 58.08 |
| Ours | 2.6 | 4.0GB | 26.0 tok/s | 60.69 |

Table 2: Effect of different codeword selection method. We report the accuracy result of LLaVA-7B model on ScienceQA dataset.

Table 3: BitWidth, Memory usage, Inference speed, and Accuracy for LLaVA-v1.5-7B on ScienceQA dataset, used to evaluate the efficiency.

| | Method | BitWidth | Subject | | | Context Modality | | | Average |
|---|--------|----------|-----|-----|-----|-----|-----|----|---------|
| | | | NAT | SOC | LAN | TXT | IMG | NO | |
| LLaVA-7B | FP | 16 bit | 65.63 | 70.64 | 68.45 | 66.18 | 65.49 | 67.46 | 67.41 |
| | GPTQ | 3 bit | 56.97 | 58.83 | 54.82 | 57.14 | 58.70 | 54.22 | 56.80 |
| | AQLM | 2.6 bit | 56.84 | 60.85 | 58.36 | 57.58 | 57.86 | 57.14 | 58.08 |
| | SparseCodeQ | 2.6 bit | 59.72 | 60.85 | 62.55 | 59.92 | 58.21 | 61.46 | 60.69 |
| | GPTQ | 2 bit | 3.42 | 2.47 | 4.00 | 3.37 | 3.47 | 3.69 | 3.37 |
| | AQLM | 2.2 bit | 54.71 | 56.47 | 59.00 | 55.38 | 55.28 | 56.17 | 56.19 |
| | SparseCodeQ | 2.2 bit | 59.24 | 61.98 | 59.09 | 59.38 | 59.49 | 58.89 | 59.77 |
| LLaVA-13B | FP | 16 bit | 70.29 | 76.94 | 71.90 | 69.99 | 70.15 | 72.82 | 72.10 |
| | GPTQ | 3 bit | 64.83 | 64.79 | 63.45 | 65.05 | 61.68 | 64.39 | 64.47 |
| | AQLM | 2.6 bit | 67.72 | 68.62 | 64.73 | 66.23 | 67.63 | 65.99 | 67.13 |
| | SparseCodeQ | 2.6bit | 69.40 | 72.55 | 68.81 | 68.23 | 67.77 | 69.47 | 69.91 |
| | GPTQ | 2 bit | 7.28 | 3.26 | 14.27 | 7.38 | 2.63 | 13.31 | 8.25 |
| | AQLM | 2.2 bit | 65.59 | 67.94 | 64.73 | 65.15 | 64.45 | 64.88 | 65.85 |
| | SparseCodeQ | 2.2 bit | 68.07 | 69.29 | 65.91 | 67.06 | 65.64 | 66.69 | 67.77 |

Table 4: Comparisons with the state-of-the-arts quantization methods for LLaVA-v1.5-7B and LLaVA-v1.5-13B models. Accuracy of visual question-answering task on ScienceQA dataset have been presented. Question classes: NAT = natural science, SOC = social science, LAN = language science, TXT = text context, IMG = image context, NO = no context.

selection strategies on model performance by constructing several variants of our approach. These include selecting codewords through random selection, Orthogonal Matching Pursuit (OMP) (Tropp & Gilbert, 2007), and our proposed HCS method. Random selection performs the worst as the chosen codeword combinations fail to represent the weight matrix effectively. OMP improves upon this by selecting codewords that approximate the original weights, though it does not explicitly minimize output error. In contrast, our HCS method adaptively selects the optimal sparse linear combination of codewords to reduce the output error, leading to improved accuracy.

## 4.2 COMPARISON WITH THE STATE-OF-THE-ART METHODS

In this section, we compare our results with state-of-the-art quantization methods: GPTQ (Frantar et al., 2022) and AQLM (Egiazarian et al., 2024). To conduct the baseline method for LVLMs, we introduce AQLM with randomly assigned codeword number for sparse coding quantization and leverage OMP for codeword selection, maintaining consistent bitwidth setting between AQLM and SparseCodeQ for fair comparison and demonstrating the superiority of salience-driven sparse coding quantization. The answering accuracy of the baseline methods is obtained by implementing the officially released code.

Table 4 presents the comparison of top-1 accuracy of different quantization methods across various LVLMs architectures including LLaVA-v1.5-7B and 13B, where bitwidths of weights for quantized layers are set from 2.2 to 2.6. GPTQ updates the quantization function using the second-order information to mitigate the quantization errors, while accumulating quantization errors due to the severely limited range of weight representation in scalar quantization methods, leading to significant degradation under extreme low bitwidth such as 2 bits. AQLM represents weight groups using full-precision codebooks with linear combinations to effectively reduce quantization errors and maintain model performance under low bitwidth constraints. However, ignoring the optimal sparse linear

| | Method | BitWidth | GQA | VizWiz | VQA$^{v2}$ | MMB | MME | VQA$^T$ | POPE |
|---|---|---|---|---|---|---|---|---|---|
| LLaVA-7B | FP | - | 61.96 | 49.87 | 78.50 | 64.30 | 1508.24 | 58.22 | 85.90 |
| | GPTQ | 3bit | 57.13 | 50.27 | 75.77 | 58.24 | 1406.54 | 52.97 | 83.64 |
| | AQLM | 2.6bit | 57.49 | 51.24 | 76.13 | 58.54 | 1349.24 | 52.67 | 83.83 |
| | SparseCodeQ | 2.6bit | 58.52 | 52.75 | 77.54 | 59.70 | 1410.36 | 54.46 | 85.41 |
| | GPTQ | 2bit | 8.40 | 2.64 | 14.40 | 3.18 | 289.98 | 2.18 | 51.62 |
| | AQLM | 2.2bit | 57.10 | 50.76 | 75.67 | 58.02 | 1302.67 | 52.04 | 83.64 |
| | SparseCodeQ | 2.2bit | 58.36 | 52.53 | 77.36 | 59.35 | 1400.33 | 53.05 | 84.76 |
| LLaVA-13B | FP | - | 63.27 | 53.63 | 79.94 | 67.47 | 1527.68 | 61.22 | 85.90 |
| | GPTQ | 3bit | 61.94 | 55.55 | 78.66 | 61.17 | 1429.91 | 57.19 | 84.46 |
| | AQLM | 2.6bit | 61.89 | 55.87 | 78.43 | 65.89 | 1458.07 | 58.59 | 84.70 |
| | SparseCodeQ | 2.6bit | 62.41 | 56.54 | 79.36 | 66.40 | 1504.67 | 59.81 | 85.76 |
| | GPTQ | 2bit | 10.13 | 0.28 | 11.96 | 5.15 | 264.04 | 4.80 | 55.46 |
| | AQLM | 2.2bit | 61.54 | 54.53 | 78.01 | 65.46 | 1430.83 | 58.45 | 84.36 |
| | SparseCodeQ | 2.2bit | 62.17 | 56.32 | 79.24 | 65.97 | 1478.36 | 59.36 | 85.36 |

Table 5: Comparisons with the SOTA quantization methods for LLaVA-v1.5 models on VQA datasets across bitwidth setting. "FP" stands for full-precision model and the performance of full-precision models in downstream datasets is provided as baseline. MMB: MMBench; VQA$^T$: TextVQA.

combination of codewords leads to suboptimal codeword selection and potential overfitting to the calibration set. On the other hand, our SparseCodeQ adaptively assigns codewords to each weight while hierarchically searching the optimal sparse linear combination of codewords, which significantly minimizes quantization errors and ensures optimal weight representation, and further optimizes the visual encoder to concentrate salience distribution for fine-grained compression while reducing search costs and further improve the model performance. As a result, our method outperforms AQLM by 3.85 (59.77 vs. 56.19) for answering accuracy in ScienceQA dataset under 2.2 bits in LLaVA-7B model. The superiority of our method becomes even more pronounced for 2.2-bit LVLMs, where quantization errors and accurate weight group representation play a more crucial role in low-capacity networks. Figure 3 illustrates the average answering accuracy in ScienceQA dataset for different quantization methods with the model size. Our SparseCodeQ reserve the optimal generation performance, while compressing the model size by 5.58x (4.33GB vs. 24.19GB) than 13B full-percision models. Furthermore, to evaluate the practical memory efficiency and inference acceleration achieved by our quantization method, we deployed the quantized 2.6-bit LLaVA-v1.5-7B model on a single NVIDIA RTX 3090 GPU shown in Table 3, where our SparseCodeQ demonstrates remarkable reduction in memory consumption for 3.60x (4.0GB vs. 14.4GB) and accelerate the inference speed for 1.31x (26.0 token/s vs. 19.8 token/s). Additionally, we evaluate our method on 7 visual question-answering (VQA) datasets using different models to verify the generalization capability. Table 5 presents the accuracy or answering scores of various quantization methods for LVLMs across multiple VQA benchmarks. Our method consistently achieves the highest accuracy across different datasets, demonstrating its robust performance across diverse downstream tasks.

## 5 CONCLUSION

In this paper, we have presented an extreme sparse coding quantization framework of 2-bit large vision-language models for efficient multimodal reasoning. Different from conventional codebook-based quantization methods which ignore the significant variance of weight salience and assign the same codeword number to all weights, we adaptively select the optimal sparse linear combinations of codewords for each weight to mitigate the performance degradation with negligible complexity overhead. Salience-driven codeword allocation and hierarchically codeword searching efficiently optimize the sparse representation. The high-level candidate search selects codeword subsets with minimal quantization errors while the low-level subset refinement discovers the fine-grained codeword combination. Optimization of the visual encoder concentrates the weight salience distribution to reduce computational overhead due to the decreased codewords for aggregated salient weights. Extensive experiments demonstrate that our methods outperform the state-of-the-art quantization methods across various multi-modal architectures.

**Reproducibility statement:** We provide detailed methodological descriptions and implementation details in the main text and supplementary material to support reproducibility. We further commit that our code will be publicly released upon acceptance of this paper.

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

## A  IMPLEMENTATION DETAILS

In this work, we leverage a large vision-language framework to perform model quantization on LLaVA (Liu et al., 2024a), utilizing its pre-trained weights to support multimodal question-answering tasks. We deploy our SparseCodeQ for 8 multimodal answer reasoning datasets including Sci-enceQA (Lu et al., 2022), GQA (Hudson & Manning, 2019), VizWiz (Gurari et al., 2018), VQA-v2 (Goyal et al., 2017), MM-Bench (Liu et al., 2024c), MME (Fu et al., 2024), TextVQA (Singh et al., 2019) and POPE (Li et al., 2023b). Our quantization approach targets various quality-efficiency trade-offs by setting the bitwidth of quantized weights to 2.6 bits and 2.2 bits in separate experimental configurations.

In practice, instead of splitting the weight matrix into rows, we divide the weight matrix $\mathbf{W}$ into a collection of smaller weight groups by partitioning each row of weights into groups of $g$ consecutive elements in AQLM (Egiazarian et al., 2024), denoted as $\{w_i\}_{i=1}^{G}$. This allows each group of weights to be mapped to a compact codeword from a codebook rather than quantizing the entire weight matrix at once, where $w_i \in \mathbb{R}^{g \times 1}$ is the $i$-th weight group and $G$ represents the total number of these groups. By partitioning high-dimensional weight tensors into smaller groups, we effectively reduce both codebook size and search complexity while maintaining high quantization accuracy through delicately designed quantization function. For the 2.6-bit configuration, we use a codebook of size $K = 2^{16}$ with a group size $g = 8$, allowing for a higher precision in weight representation. For the 2.2-bit configuration, we employ a slightly smaller codebook of size $K = 2^{15}$ while keeping the group size constant at $g = 8$, aiming for improved computational efficiency. We set the maximum codeword number to 3 for all weights to achieve satisfying trade-offs between the discretization errors and the search cost. We followed the initialization of the quantization function parameters in AQLM for the baseline methods and our SparseCodeQ. The quantization function parameters were updated for 10 epochs in searching process, and the acquired discretization function was directly employed for multimodal question answering.

To further optimize the model's vision encoder, we use the CC-3M Concept-balanced dataset (Liu et al., 2024a), which consists of 595K images with rich and balanced visual concepts to ensure diverse visual inputs. Additionally, we conduct a calibration dataset specifically for quantizing and fine-tuning the quantized model. This dataset includes 1024 text-only sequences sampled from the RedPajama dataset (Computer, 2023), each containing 4096 tokens, and 64 image-text sequences from the CC-3M Concept-balanced dataset. To match the token length, we extend the text portion of each image-text sequence to 4096 tokens using GPT-4 (Achiam et al., 2023). This calibration setup ensures the quantized model is well-adapted to both text-only and multimodal inputs across sequence lengths and varied visual contexts. During fine-tuning, we adopt the Adam optimizer with a learning rate of 0.0001 to optimize the model parameters effectively.

## B  PERFORMANCE ON MORE BASELINE METHODS AND DIFFERENT LVLMs ARCHITECTURES

We first conduct experiments on more baseline methods includes AWQ (Lin et al., 2023) and QTIP (Tseng et al., 2024b) in LLaVA-v1.5-7B model on ScienceQA dataset under different bitwidth setting. AWQ is a post-training quantization (PTQ) method that identifies the optimal scale to protect the salient weight channels and decrease the quantization errors for weight quantization. However, it suffers from unbearable quantization errors and accuracy drop under extreme low-bitwidth due to constrained numerical precision with severe information loss. QTIP improves codebook search efficiency through trellis-coded quantization and a hardware-efficient design based on Gaussian codes, while incurring compromise quantization accuracy due to the absence of additional information injection for feature representation. Moreover, neither method addresses the distributional discrepancies between vision and language modalities in multi-modal models, resulting in coarse quantization function design with severe quantization errors. On the contrary, our framework achieves optimal codeword combination for weight feature representation through adaptive salience-driven codeword allocation and hierarchical codeword selection methods. Furthermore, we optimize vision encoder to concentrate the data distribution within vision modality for better consistency of vision-language feature spaces. As a result, our method outperforms QTIP by 2.76 (60.69 vs. 57.93), with the advantage becoming even more pronounced under 2.2-bit quantization where accurate weight representation plays a more critical role in low-capacity networks.

| | Method | BitWidth | Subject | | | Context Modality | | | Average |
|---|---|---|---|---|---|---|---|---|---|
| | | | NAT | SOC | LAN | TXT | IMG | NO | |
| LLaVA-7B | FP | 16 bit | 65.63 | 70.64 | 68.45 | 66.18 | 65.49 | 67.46 | 67.41 |
| | AWQ | 3bit | 58.68 | 60.49 | 62.82 | 58.91 | 57.26 | 60.75 | 60.13 |
| | QTIP | 2.6 bit | 56.41 | 60.17 | 57.96 | 56.48 | 55.36 | 59.32 | 57.93 |
| | SparseCodeQ | 2.6 bit | 59.72 | 60.85 | 62.55 | 59.92 | 58.21 | 61.46 | 60.69 |
| | AWQ | 2bit | 2.67 | 5.32 | 4.76 | 3.76 | 4.17 | 4.96 | 3.74 |
| | QTIP | 2.2 bit | 55.23 | 57.53 | 58.63 | 57.34 | 56.03 | 58.01 | 56.73 |
| | SparseCodeQ | 2.2 bit | 59.24 | 61.98 | 59.09 | 59.38 | 59.49 | 58.89 | 59.77 |
| TinyLLaVA-3B | FP | 16 bit | 78.86 | 76.94 | 70.91 | 78.25 | 73.97 | 72.47 | 76.40 |
| | AQLM | 3.0 bit | 55.15 | 52.90 | 53.10 | 56.35 | 54.60 | 51.76 | 54.12 |
| | SparseCodeQ | 3.0 bit | 61.50 | 57.98 | 57.34 | 61.51 | 59.66 | 56.71 | 59.66 |
| | AQLM | 2.2 bit | 42.27 | 28.88 | 48.33 | 42.72 | 38.26 | 45.00 | 41.81 |
| | SparseCodeQ | 2.2 bit | 44.77 | 30.55 | 50.08 | 45.38 | 40.04 | 47.04 | 42.77 |
| Qwen2.5-VL-3B | FP | 16 bit | 82.37 | 81.78 | 76.55 | 80.55 | 80.42 | 77.91 | 80.74 |
| | AQLM | 3.0 bit | 55.73 | 62.54 | 53.62 | 53.27 | 59.17 | 58.96 | 58.37 |
| | SparseCodeQ | 3.0 bit | 59.68 | 67.49 | 62.82 | 58.74 | 63.58 | 60.51 | 62.13 |
| | AQLM | 2.2 bit | 47.38 | 38.13 | 38.82 | 48.18 | 44.58 | 40.67 | 43.81 |
| | SparseCodeQ | 2.2 bit | 52.22 | 49.41 | 49.27 | 53.17 | 51.23 | 45.41 | 50.91 |

Table 6: Comparisons with the state-of-the-arts quantization methods for TinyLLaVA-3B and Qwen2.5-VL-3B model. Accuracy of visual question-answering task on ScienceQA dataset have been presented. Question classes: NAT = natural science, SOC = social science, LAN = language science, TXT = text context, IMG = image context, NO = no context.

We further explore our quantization results with AQLM on different large vision-language models (LVLMs) architectures, using TinyLLaVA-Qwen2.5-3B (Zhou et al., 2024) and Qwen2.5-VL-3B (Bai et al., 2025) as a representative case study. Table 6 presents the top-1 accuracy results on ScienceQA dataset across various bitwidth settings. SparseCodeQ designed in LLaVA-like architectures can be effectively adapted to window-attention based VLMs with stronger connection between the image and text due to the consistent core mechanism of window-attention and the robust multi-modal alignment capabilities pre-trained on large-scale vision-language pairs. Since Qwen2.5-VL is a window-attention based VLM, exploiting salience-driven codeword allocation and hierarchical codeword selection for optimal codeword combination remain suitable. Notably, the 3B-scale models becomes more sensitive for quantization than 7B-scale models, and AQLM encounters significant accuracy drop due to suboptimal codeword selection. On the contrary, our SparseCodeQ construct optimal sparse linear combination of codewords for weight group representation with reduced quantization errors. As a result, our method outperforms AQLM by 5.54 (59.66 vs. 54.12) for answering accuracy in ScienceQA dataset under 3 bits in TinyLLaVA-3B model. These results underscore the robustness and generalizability of our approach across different tasks, model architectures and datasets, demonstrating its effectiveness in diverse scenarios.

## C  LIMITATION

Although codebook-based quantization methods significantly reduce model size and memory usage for large models, achieving outstanding performance compared to post-training quantization (PTQ) under extreme bitwidth, they encounter modest increase for latency due to increased computational complexity in matrix multiplication operations required for weight reconstruction during inference.

## D  LLM USAGE STATEMENT

Large Language Models (LLMs) were used for language polishing in this work. The authors take full responsibility for the content and results presented.

