# OpenReview forum: "SparseCodeQ: Extreme Sparse Coding Quantization for Large Vision-Language Models"
_ICLR.cc/2026/Conference — ICLR 2026 Conference Withdrawn Submission_

### Official Review · Reviewer_3u7Q · 2025-11-01

**Soundness:** 3
**Presentation:** 3
**Contribution:** 3
**Rating:** 6
**Confidence:** 3

**Summary:**

This paper proposes SparseCodeQ. To address the discretization error problem in 2-bit quantization of LVLMs, the method dynamically allocates the number of codebooks based on weight salience and optimizes the visual encoder to concentrate the salience distribution. The authors claim that the method achieves a 5.58× reduction in the size of the 13B LLaVA model.

**Strengths:**

The paper innovatively integrates sparse coding with quantization, dynamically allocating codebooks to address the issue that traditional methods overlook the variance in weight salience. It features clear writing and conducts extensive experiments covering multiple datasets and model architectures.

**Weaknesses:**

The paper has insufficient theoretical analysis on the cross-image similarity of weight salience, merely mentions multimodal extension without providing validation, and the theoretical analysis of the method is relatively simplistic. Additionally, it lacks hardware-level validation and fails to evaluate the storage and computational overhead during actual deployment.

**Questions:**

1. Increase the analysis of different weight salience evaluation methods: what methods can be used for salience-based codebook allocation, what impacts do these methods have on performance, and why do these impacts occur? Sufficient analysis is required to reveal the reasons for performance improvement, rather than merely increasing the number of codebooks.
2. What is the specific computational overhead of hierarchical codeword selection? It is necessary to analyze the additional computational load introduced by the method. Additionally, attempts can be made to propose methods for further optimizing its efficiency to provide insights for future work.
3. I do not fully understand the basis for setting the weights of the entropy minimization objective in visual encoder optimization. It is necessary to add ablation experiments on hyperparameters.

---

### Official Review · Reviewer_SXev · 2025-11-01

**Soundness:** 2
**Presentation:** 2
**Contribution:** 2
**Rating:** 2
**Confidence:** 3

**Summary:**

The paper proposes a sparse-coding quantization scheme for low-bit LVLMs. It allocates variable codewords per group using Hessian-based salience, performs hierarchical codeword selection, and regularizes the vision encoder with a Hessian-entropy term to concentrate salience. Results show accuracy gains at 2–3 bits and modest speedups.

**Strengths:**

- Targets extreme low-bit settings where many PTQ baselines fail.

- Methodically designed selection stages reduce naive combinatorial search.

- Attempts model-side distribution shaping rather than only post-hoc mapping.

**Weaknesses:**

1. Proxy–objective gap. Allocation is driven by diagonal Hessian of layer reconstruction on calibration data, while evaluation is end-task VQA accuracy. The causal link between concentrated Hessian mass in the vision encoder and reduced end-to-end quantization error through the language stack is asserted, not established.

2. Hidden compute and bandwidth overhead. Multiple codewords per group imply sparse linear combinations at inference, increasing index fetches and accumulations. Despite large memory savings, the reported latency gain is small, indicating arithmetic intensity and cache residency may be worse than stated. No kernel-level analysis (index width, packing, FLOP/byte, cache hit rates) is provided.

3. Calibration sensitivity and scaling risk. Hessian estimates and hierarchical selection depend on limited calibration data. The approach is likely sample- and distribution-sensitive, but there are no stress tests under prompt-length scaling, domain shift, or smaller calibration budgets. Codebook and index streams also introduce rate overhead that is not fully accounted for in bytes-per-parameter.

**Questions:**

N/A.

---

### Official Review · Reviewer_bsUs · 2025-11-04

**Soundness:** 2
**Presentation:** 2
**Contribution:** 3
**Rating:** 4
**Confidence:** 3

**Summary:**

This paper introduces SparseCodeQ, a 2-bit quantization framework for MLLMs. It solves the problem of conventional methods which assign a fixed number of codewords to all weights, ignoring their different importance. SparseCodeQ, instead, flexibly assigns an optimal sparse linear combination of codewords to each weight based on its salience, which is evaluated using second-order information. It uses a hierarchical search to efficiently find the best codeword combination. Experiments show this method achieves a 5.58x reduction in model size while outperforming state-of-the-art methods.

**Strengths:**

1. Adaptive Quantization Based on Salience. A primary strength is its novel approach. Unlike conventional codebook methods that assign the same number of codewords to all weights , SparseCodeQ is the first framework to use sparse coding principles for model compression. It adaptively assigns an optimal _sparse linear combination_ of codewords to each weight based on its salience. This allows for a more fine-grained representation of important parameters, which significantly mitigates performance degradation.
2. Superior Performance at Extreme Compression. The method demonstrates state-of-the-art results, especially at very low bitrates (e.g., 2.2 bits). Experiments on the 13B LLaVA model show SparseCodeQ achieves a 5.58x reduction in model size while outperforming state-of-the-art quantization methods by 2.78 in performance. It also shows practical efficiency gains, achieving a 3.6x memory reduction and 1.3x inference acceleration on a 7B model.

**Weaknesses:**

Evaluation is focused on short-output tasks. The paper's experiments, while extensive, are heavily focused on VQA-style benchmarks (e.g., ScienceQA, VQA-v2, GQA). These tasks typically require short, often single-word or option-based, answers. This is a limitation because the impact of extreme quantization on long-form generative tasks (like detailed image description or open-ended multimodal dialogue) is not evaluated. Quantization errors can accumulate differently and cause more significant quality degradation in longer, coherent text sequences. The model's performance in these more open-ended scenarios remains unverified.

**Questions:**

How would SparseCodeQ perform on even larger-scale MLLMs, and does its effectiveness scale?

---

### Official Review · Reviewer_XcLx · 2025-11-05

**Soundness:** 3
**Presentation:** 3
**Contribution:** 3
**Rating:** 4
**Confidence:** 5

**Summary:**

This paper proposed a novel extreme sparse coding quantization framework for ultra-low bit large vision-language models targets on multimodal reasoning. Authors design a flexible codeword combination for each wright based on weight salience, which involves selecting the number of codewords based on salience evaluation and hierarchical codeword selection to search an appropriate codeword combination with mininal quantization errors to reducing the large codebook. Experimental results prove the effectiveness of the proposed methods mainly on LLaVA models.

**Strengths:**

1. The writing and logic are clear and reasonable, which are easy to follow.
2. Pushing the limit of quantized LVLMs into 2-bit is contributing and the experimental results also prove the probability of applying the proposed methods into industry or for further academic research.
3. Figures are clear and informative enough for me to follow.

**Weaknesses:**

1. Section 3.2 is derived and widely used in previous arts, like HAWQ, BRECQ, etc. which can be omitted or moved into supplementary for better writing logic. Also, this part should not be treated as a contribution.
2. In "High-level candidate search" of Section 3.3, how to define the size of the potential codeword subsets and how does the size effect the final performance and quantization efficiency of the proposed methods? Authors should discuss more with experimental or theoretical analysis.
3. The condition and design of codewords can be different between text embedding and vision encoders since LVLMs take both text and image tokens as inputs, while these two modals can be very different in both distribution range and outliers. How to address this phenomenon with the proposed method? In other words, authors fail to discuss or solve the challenges raised by the modality.

**Questions:**

See weaknesses.

---

### Note · Authors · 2025-11-14

I have read and agree with the venue's withdrawal policy on behalf of myself and my co-authors.